# CTIVA: Censored time interval variable analysis

**Insoo Kim**[1], **Junhee Seok**[1], **Yoojoong Kim**[2]*

**1** School of Electrical Engineering, Korea University, Seoul, Republic of Korea, **2** School of Computer Science and Information Engineering, The Catholic University of Korea, Bucheon, Republic of Korea

* yoojoongkim@catholic.ac.kr

**Data Availability Statement:** The National Sample Cohort Demo (NSCD) used in the current study provided from the third party; National Health Insurance Service (NHISS) is freely available for domestic users, which are citizens of South Korea, and available for foreigners with certain amount of

## Abstract

Traditionally, datasets with multiple censored time-to-events have not been utilized in multivariate analysis because of their high level of complexity. In this paper, we propose the Censored Time Interval Analysis (CTIVA) method to address this issue. It estimates the joint probability distribution of actual event times in the censored dataset by implementing a statistical probability density estimation technique on the dataset. Based on the acquired event time, CTIVA investigates variables correlated with the interval time of events via statistical tests. The proposed method handles both categorical and continuous variables simultaneously—thus, it is suitable for application on real-world censored time-to-event datasets, which include both categorical and continuous variables. CTIVA outperforms traditional censored time-to-event data handling methods by 5% on simulation data. The average area under the curve (AUC) of the proposed method on the simulation dataset exceeds 0.9 under various conditions. Further, CTIVA yields novel results on *National Sample Cohort Demo* (NSCD) and proteasome inhibitor bortezomib dataset, a real-world censored time-to-event dataset of medical history of beneficiaries provided by the *National Health Insurance Sharing Service* (NHISS) and National Center for Biotechnology Information (NCBI). We believe that the development of CTIVA is a milestone in the investigation of variables correlated with interval time of events in presence of censoring.

## Introduction

Discovering significant variables related to particular phenomena via multivariate analysis is an important task in several academic fields. In particular, identifying causal relationships between clinical events and gene expression analysis is a major problem in bioinformatics. However, clinical events are censored in most gene expression datasets, hindering the evaluation of relationships between gene expression data and clinical events. Although several methods have been proposed to handle the censored datasets [1–5], handling the samples with multiple censored events remains a complicated task [6].

Development of a method capable of handling multiple censored datasets could provide an effective solution to several problems in multiple fields. For instance, it would be useful for the identification of a gene related to the survival times of patients with carcinoma—the gene can

download fee. The NSCD datasets generated and/or analyzed during the current study are available in the [https://nhiss.nhis.or.kr/bd/ab/bdaba022eng.do] repository. Foreign investigators approved for the data sharing from the health insurance corporation in purpose of policy and academic research will pay the fee for data. The simulation sample generation code and CTIVA conducting code with the proteasome inhibitor dataset is provided in the URL below [https://github.com/Insoo-K/CTIVA].

**Funding:** This research was supported by a grant of the Basic Science Research Program through the National Research Foundation of Korea funded by the Ministry of Education (NRF-2021R1I1A01044255) and was supported by the Research Fund, 2023 of The Catholic University of Korea. The funders had no role in study design, data collection and analysis, decision to publish, or preparation of the manuscript.

**Competing interests:** The authors have declared that no competing interests exist.

be denoted by a variable, the detection of carcinoma can be considered to be event 1, and detection of death can be considered to be event 2. In such a case, data corresponding to both event 1 and event 2 may be censored in various cases. Although, there exist survival analysis models that handle multiple events such as competing risk regression model, it only handles the correlated events [7]. Therefore, method that handles multiple independent censored events, should be further investigated.

Although previous studies handling multiple independent censored events presence, most of the existing works only handle the problems with continuous variables [8]. Consequently, the importance of methods that manage multiple censored datasets, accommodating various types of variables, including categorical variables and continuous variables has increased. However, despite its apparent utility, the complexity arising from multiple censoring has hindered its further development.

Traditionally, two approaches have been developed to handle censored data—statistical analysis methods [1, 2, 9] and deep learning methods [10–13] are two dominant solutions. Although deep neural network-based methods have been demonstrated to offer effective solutions on censored datasets, the inaccessibility and scarcity of multiple censored datasets limit the implementation of deep learning-based methods as a viable solution [14–16]. In this study, we propose a statistical density estimation method, called the censored-data time interval variable analysis (CTIVA), based on multiple censored data analysis.

The proposed method handles the multiple censored data through estimating the density function of the dataset and reconstructs the censored events through the estimated density functions. These statistical approaches are aimed to handle the inaccessibility and scarcity of the most censored datasets. Moreover, CTIVA demonstrates practical advantages compared to previous methods. Our proposed method addresses challenges posed by diverse variable types, a facet not adequately tackled by prior studies. Specifically, the proposed method could be implemented in clinical trial and symptoms which could be considered as categorical variables. Thus, the proposed CTIVA has extended the solution of multivariate censored time-to-event dataset to more general fields.

## Materials and methods

### Censored time-to-event data analysis and data generation

The proposed CTIVA method detects variables correlated with the interval time of events in the presence of censoring of the event. It is a generalized version of censored time-to-event data-handling methods, which are usually utilized for genomic data analysis [6]. Formally, for each sample $i$, let us denote the true occurrence times of events 1 and 2 by $T_{i1}$ and $T_{i2}$, respectively. Additionally, the censored times of events 1 and 2 are denoted by $C_{i1}$ and $C_{i2}$, respectively. In the presence of censoring, the censoring time points, $V_{i1}$ and $V_{i2}$, and censoring indicators, $\Delta_{i1}$ and $\Delta_{i2}$, are observed instead of $T_{i1}$, $T_{i2}$, $C_{i1}$, and $C_{i2}$. The censoring time points and indicators are defined as follows: $V_{i1} = \min(T_{i1}, C_{i1})$, $V_{i2} = \min(T_{i2}, C_{i2})$, $\Delta_{i1} = I(T_{i1} \leq C_{i1})$, and $\Delta_{i2} = I(T_{i2} \leq C_{i2})$ where $I(\cdot)$ denotes an indicator function. CTIVA is proposed to detect variables that are significantly associated with the true interval time, $T_2 - T_1$, derived based on the observation data $\{V_1, \Delta_1, V_2, \Delta_2\}$ of $n$ samples.

To acquire a distinctive dataset for the experiment, simulation data are generated using the following procedure. The two actual event times, $T_1$ and $T_2$, are sampled using a chosen probability density function. Subsequently, the two censoring time points, $C_1$ and $C_2$, are sampled using the similar but independent probability density functions from which the actual event times are sampled.

To handle the censored time-to-event data, the proposed CTIVA estimates the distribution of the events statistically using the joint distribution, calculates the conditional expectations of $T_2 - T_1$ based on the observed $\{V_1, \Delta_1, V_2, \Delta_2\}y$ using Monte Carlo Simulation, and, finally, identifies significant variables based on statistical tests. The detailed procedure of CTIVA is described below.

## Joint probability density estimation

Owing to the presence of censoring in the observation dataset, most of the true interval time values cannot be acquired from it. Hence, the $f_{T_1, T_2}$, the joint probability distribution of $T_1$ and $T_2$ is estimated from the observed $\{V_1, \Delta_1, V_2, \Delta_2\}$ by implementing the multivariate survival analysis of the optional Polya tree (OPT) Bayesian estimation [17, 18]. Although a general $p$-dimensional multivariate problem is handled in the previous studies, the problem is simplified as a two-dimensional bivariate problem in CTIVA for computational efficiency. The simplified method implemented in CTIVA is explained below, the detailed procedure of the original method is described in Seok *et al.* [18].

CTIVA utilizes an OPT to estimate the joint distribution. The OPT is characterized by a likelihood function $\Phi(A)$ for region $A$ in a sample space $\Omega$. $\Phi(A)$ is calculated recursively using $\Phi(A_{11})$, $\Phi(A_{12})$, $\Phi(A_{21})$, $\Phi(A_{22})$ where $A_{ij}$ is defined to be the $j$-th subregion of partition $A$, which is split by the center point of the $T_i$ axis. Formally, $\Phi(A)$ is calculated through the equation denoted below.

$$\Phi(A) = \frac{1}{2}\Phi_0(A) + \frac{1}{4}\sum_{i=1}^{2} \frac{B(N(A_{i1}) + 0.5, N(A_{i2}) + 0.5)}{B(0.5, 0.5)} \Phi(A_{i1})\Phi(A_{i2}) \tag{1}$$

The $\Phi_0(A)$ is a milestone likelihood value, assuming all sample points in $A$ to follow the uniform distribution, $B(\cdot)$ denotes a beta function, and $N(A)$ denotes the number of samples in region $A$. If $N(A) < 2$, $\Phi(A) = \Phi_0(A)$. Through the multiple recursive calculation and binary splitting, the values $\Phi(A)$ corresponding to all subregions of $\Omega$ can be obtained. Based on the obtained $\Phi(A)$ values, the joint distribution of $T_1$ and $T_2$ is calculated via CTIVA by following the steps below.

For an arbitrary subregion $A$, if $\frac{1}{2}\Phi_0(A) > +\frac{1}{4}\sum_{i=1}^{2}\frac{B(N(A_{i1})+0.5,N(A_{i2})+0.5)}{B(0.5,0.5)} \Phi(A_{i1})\Phi(A_{i2})$, then $A$ is considered to follow a uniform distribution. The probability density of $A$, the subregion following the uniform distribution, can be calculated as $\frac{N(A)}{n|A|}$ where $n$ is the total number of observed samples and $|A|$ is the area of the $A$. If $\frac{1}{2}\Phi_0(A) < +\frac{1}{4}\sum_{i=1}^{2}\frac{B(N(A_{i1})+0.5,N(A_{i2})+0.5)}{B(0.5,0.5)} \Phi(A_{i1})\Phi(A_{i2})$, the subregion $A$ is considered to follow a non-uniform distribution and is divided into partitions for the calculation—into $A_{11}$ and $A_{12}$ if $B(N(A_{11})+0.5, N(A_{12})+0.5)\Phi(A_{11})\Phi(A_{12}) > B(N(A_{21})+0.5, N(A_{22})+0.5)\Phi(A_{21})\Phi(A_{22})$ and into $A_{21}$ and $A_{22}$ otherwise.

The CTIVA performs the proposed task recursively until all partitions of the $\Omega$ are considered to follow the uniform distribution. Using the following process, the probability density of each partition is obtained based on the numbers of samples in each region.

However, the numbers of samples in region $A$, $N(A)$ cannot be estimated by counting the observations in the partition due to the missing observations occurred by censoring. In the presence of censoring, $N(A)$ is indirectly estimated from the joint distribution $f_{T_1, T_2}$. For a given joint distribution $f$, we define $N(A|f)$ be the estimated number of samples in $A$. Since $N(A)$ values are required for OPT calculation, $f_{T_1, T_2}$ is obtained through the equation below by substituting $N(A)$ to $N(A|f)$.

$$f_{T_1, T_2} = \text{OPT}(N(A|f)) \tag{2}$$

To solve the proposed equation above, an iterative approach where $f_{T_1,T_2}^{(i+1)} = \text{OPT}(N(A|f_{T_1,T_2}^{(i)}))$ is implemented. The proposed CTIVA method obtains the final joint distribution $f_{T_1,T_2}$ by repeating the iteration until $f_{T_1,T_2}^{(i)}$ converges. The initial distribution essential for iteration is obtained from the initial estimation of $N(A)$. $N^{(0)}(A)$ is estimated from $T_1$ and $T_2$, assuming two distributions to be independent in subregion $A$. Univariate Kaplan-Meier estimators are used for the initial estimation of $T_1$ and $T_2$ in region $A$. Finally, the initial joint probability distribution is given by $f_{T_1,T_2}^{(1)} = \text{OPT}(N(A|f_{T_1,T_2}^{(0)}))$. All of the functionally notated values suggested in the OPT algorithm are discrete values which implies the usual convergence of the algorithm.

### Time interval estimation

Using the estimated joint distribution $f_{T_1,T_2}$ obtained via the aforementioned steps, the conditional distribution of $T_1$ and $T_2$ can be calculated when the observations of sample $i$ are given. The given observation $\{V_{i1}, \Delta_{i1}, V_{i2}, \Delta_{i2}\}$, can be classified into four cases. In the first case, $\Delta_{i1} = 1$ and $\Delta_{i2} = 1$, and censoring is absent. Therefore, $T_{i1} = V_{i1}$ and $T_{i2} = V_{i2}$. The second and third cases involve single censoring. In the second case, $\Delta_{i2} = 0$, and, in the third case, $\Delta_{i1} = 0$. Thus, $T_{i1} = V_{i1}$ in the second case and $\Pr[T_{i2}|T_{i1} = V_{i1}, T_{i2} > V_{i2}]$ is obtained. On the other hand, $T_{i2} = V_{i2}$ in the third case and $\Pr[T_{i1}|T_{i1} > V_{i1}, T_{i2} = V_{i2}]$ is obtained. In the final case, censoring exists for both events. Thus, $\Delta_{i1} = 0$ and $\Delta_{i2} = 0$ in the final case, and $\Pr[T_{i2}|T_{i1} > V_{i1}, T_{i2} > V_{i2}]$ is obtained.

After estimating the density functions in the four cases, $E[T_2 - T_1|V_1, \Delta_1, V_2, \Delta_2]$ is estimated empirically via Monte Carlo Simulation because the analytical calculation of the expectation is complicated and inaccurate due to the presence of random sampling in the OPT algorithm when the censoring exists. To reduce the influence of the random sampling in the OPT algorithm and save the computational resources, Monte Carlo Simulation is implemented to obtain the empirical expectation. From the obtained conditional distributions of $T_1$ and $T_2$, the pairs $(T_1, T_2)$ are randomly sampled for the calculation. The empirical expectation is calculated as the mean of the interval $T_2 - T_1$ of the generated pair.

Let $x_{ij}$ be the observation of $j$-th variable, $x_j$, in sample $i$ and $y_i$ be the expected interval time, $E[T_{i2} - T_{i1}|V_{i1}, \Delta_{i1}, V_{i2}, \Delta_{i2}]$, obtained via Monte Carlo simulation. Based on the pairs, $(x_{ij}, y_i)$ for $i = 1, 2, \ldots, n$, the statistical relationships between the variables, $x_j$ and $T_2 - T_1$, can be estimated via several statistical estimation methods, such as analysis of variance (ANOVA), permutation tests, and rank correlation tests.

## Results

### Simulation settings

The statistical and stochastic approach of CTIVA in handling censored datasets has been demonstrated to be effective in previous studies. However, the direct demonstration of the novelty of the proposed method in real-world problems is difficult owing to poor data accessibility of major censored datasets. Therefore, we first investigate the novelty of the proposed CTIVA method using simulated examples before we testing it on real-world problems.

Initially, a bivariate censored time-to-event dataset comprising 1,000 categorical variables is randomly generated. The data are randomly sampled from three different probability density functions—additive exponential distribution, log-normal distribution, and the Clayton-Oakes model [19]. Each dataset consists of 500 generated samples and the experiment is repeated 100 times using the identical generation procedure. A detailed description of the probability density function is provided in Table 1.

**Table 1. Detailed analysis of data generation procedure.**

| | T | C |
|---|---|---|
| **Additive Exponential** | $T_1 \sim \exp(t)$<br>$T_2 \sim T_1 + \alpha$<br>$\alpha \sim \exp(t), \alpha \perp T_1$ | $C_1, C_2 \sim \exp(c)$<br>$C_1 \perp C_2$ |
| **Log-Normal** | $\log\begin{pmatrix} T_1 \\ T_2 \end{pmatrix} \sim N(\mu_T, \, \Sigma_T)$ | $\log\begin{pmatrix} C_1 \\ C_2 \end{pmatrix} \sim N(\mu_C, \, \Sigma_C)$ |
| **Clayton-Oakes Model** | $T_1, T_2 \sim S(t_1, t_2)$ | $C_1, C_2 \sim \exp(c)$<br>$C_1 \perp C_2$ |

The T refers to the actual event time of the generated data and C refers to the censored time point of the data. The $N(\mu_T, \Sigma_T)$ refers to the Gaussian normal distribution with $\mu_T$ as set of means and $\Sigma_T$ as set of covariances. The $S(t_1, t_2)$ denotes the bivariate survival function in Clayton-Oakes model.

Variables sampled from three different distribution have distinct characteristics. The variables sampled from additive exponential distribution have monotonic hazard function while variables sampled from log-normal distribution have non-monotonic hazard function. Unlike the first two, variables sampled from Clayton-Oakes model have inherent dependency between two event times [20].

Out of 1,000 variables, 100 variables were generated to be correlated to the interval time of event 1 and 2, other 100 variables were generated to be correlated to the occurrence time of the event 1, other 100 variables were generated to be correlated to event 2, and the remaining 700 variables were independently generated with the events. The detailed generation procedure for the dataset is summarized in Table 1. The proposed CTIVA was designed to detect the 100 variables that were correlated to the interval time among 1,000 variables. The variables were detected to be correlated according to a statistical significance investigated through the various statistical tests. Since the variable detection results depend on the threshold value, by regulating the threshold of the p-value, the Receiver Operating Characteristic (ROC) curves and Area Under Curve (AUC) values are obtained.

The proposed CTIVA method was compared with various baseline methods for the performance verification. The baseline methods include naïve methods that do not consider the presence of censoring by ignoring the censoring or only considering the sample without censoring. In specific, the naïve method that ignores censoring considers the censored time event as actual observed event. Furthermore, we have also conducted the experiment with Cox Regression Model [3], the method that is dominantly used for handling the censored time to event data.

## Simulation results in uncorrelated dataset

The effectiveness of the proposed method to solve real-world problems is demonstrated by applying it to combined variable problems, as real-world problems usually involve both categorical and continuous variables. To generate the combined variable problem, a bivariate censored time-to-event dataset comprising 500 categorical variables and 500 continuous variables is randomly generated. Time-to-events are randomly sampled from the three density functions listed in Table 1. As in the case of the dataset with 1,000 categorical variables, 100 variables that were half continuous and half categorical are observed to be highly correlated with the interval times of both events 1 and 2. Other variables exhibit similar degrees of correlation as in the case of the dataset with 1,000 categorical variables, except that half of them are continuous. The same statistical tests are performed on the combined dataset, except for linear-regression-

based p-value estimation, which is used to replace the ANOVA test for continuous variables. Both the ROC curve and AUC values are acquired on the combined dataset. The simulation is conducted using identical sample numbers and repetitions as those in the case of 1,000 categorical datasets. Although the tests implemented in the cases of categorical and continuous variables are different, the same p-value threshold is used to calculate the ROC curve and AUC value during the evaluation of the method. The underlying mathematical basis for the equality of thresholds might be insufficient—the experiments are designed to demonstrate the effectiveness of the proposed method in the case of real-world problems, which usually involve both categorical and continuous variables.

The proposed CTIVA method exhibits excellent prediction performance corresponding to both categorical and combined variables that are correlated with the interval time. The performance of the CTIVA is compared with Cox survival analysis model [1] which only considers single censored time event and two naïve methods that do not handle the censoring. The Cox model is implemented with only a single censored time events while the other event is ignored. For the two naïve methods, tests are conducted only for the samples with non-censored data in former case and censoring present in the data is simply ignored for latter case. CTIVA exhibits average AUC values of 0.93, 0.94, and 0.93 in the case of categorical variables—with ANOVA, permutation test, and rank correlation test while the Cox model and two naïve methods exhibit AUC values lower than 0.9. Additionally, the average AUC values of CTIVA in the combined variable case are 0.91, 0.91, and 0.90, which are higher than those of the compared methods. An additive exponential distribution is used as the sampling distribution in the experiment. The average AUC values of CTIVA are observed to be 4.5, 5.6, and 8.1% higher than those of the most competitive alternative considered in the experiment on categorical dataset, 5.8, 5.8, and 6.7% higher on the combined dataset. The detailed comparison results are presented in Table 2.

The box plots of the AUC values obtained via repeated simulation of categorical and combined datasets using different sampling distributions are depicted in Figs 1 and 2 respectively to demonstrate the visualized performance. The constants in Fig 2 denote the power of the additive exponential of the sample distribution. Evidently, the proposed CTIVA method outperforms traditional methods in terms of both average and ranged AUC values on simulation datasets with various environments.

Furthermore, for the cutoff p-value 0.05, the proposed CTIVA showed novel results in both sensitivity and specificity compared to other baseline methods. With the cutoff p-value 0.05 in

**Table 2. The AUC comparison results of proposed CTIVA and other baseline models in uncorrelated dataset.**

| Data Type | Test Method | ANOVA (+Linear) | Permutation Test | Rank Correlation Test |
|---|---|---|---|---|
| Categorical | CTIVA | 0.93±0.04 | 0.94±0.03 | 0.93±0.03 |
| | Cox Event 1 | 0.61±0.03 | | |
| | Cox Event 2 | 0.82±0.03 | | |
| | Ignored | 0.82±0.06 | 0.82±0.06 | 0.78±0.06 |
| | No Censor | 0.89±0.02 | 0.89±0.02 | 0.86±0.03 |
| Combined | CTIVA | 0.91±0.06 | 0.91±0.06 | 0.90±0.02 |
| | Cox Event 1 | 0.61±0.02 | | |
| | Cox Event 2 | | 0.78±0.03 | |
| | Ignored | 0.80±0.05 | 0.80±0.05 | 0.79±0.05 |
| | No Censor | 0.86±0.03 | 0.86±0.05 | 0.84±0.03 |

The dataset was sampled from an additive exponential distribution with different statistical tests.

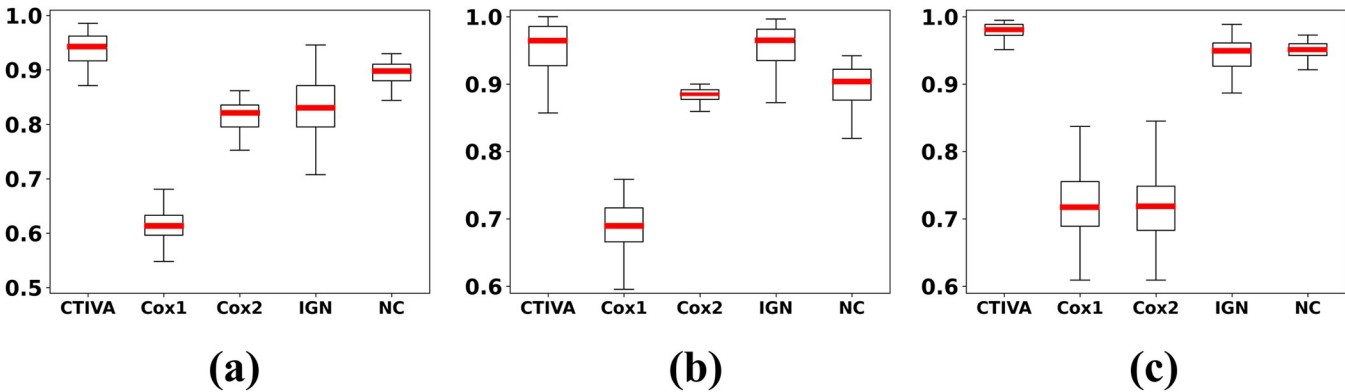

**Fig 1. Box plot of AUC in predicting categorical variables with ANOVA test.** The CTIVA indicates the proposed method while other methods are baseline methods depicted for comparison. The Cox1 and Cox2 describes the AUC results of Cox-survival analysis model each applied in censored time event 1 and 2. The IGN denotes the baseline methods that ignores the censoring and the NC denotes the baseline methods that only handles the non-censored data. The data of the experimental results were sampled from (a) an additive exponential distribution, (b) the log-normal distribution, and the (c) Clayton-Oakes model.

categorical dataset, the average sensitivity of CTIVA is 0.96 while the average specificity is 0.84. Also in combined dataset, the average sensitivity of CTIVA is 0.84 while the mean specificity is 0.73. The proposed CTIVA consistently outperformed other benchmark methods in terms of sensitivity or specificity. Additionally, the proposed method also demonstrated robust performance across varying p-value thresholds in the sensitivity analysis. The thresholds of 0.1 and 0.01 were additionally examined in the sensitivity analysis. The average sensitivity of the CTIVA is 0.89 and 0.72 in categorical and combined dataset with cutoff p-value of 0.01. The average specificity of the CTIVA is 0.89 and 0.90 in categorical and combined dataset. The detailed comparison result showing the novelty of the proposed CTIVA is provided in S1 and S2 Tables.

The average AUC values of CTIVA model are higher and the variances are lower than compared methods, which implies the stability of the proposed method in changing threshold. Moreover, the proposed method demonstrated novel results in dominant cutoff p-value compared to baseline methods. In conclusion, the suggested CTIVA method showed superior performance in both fixed cutoff value and changing threshold.

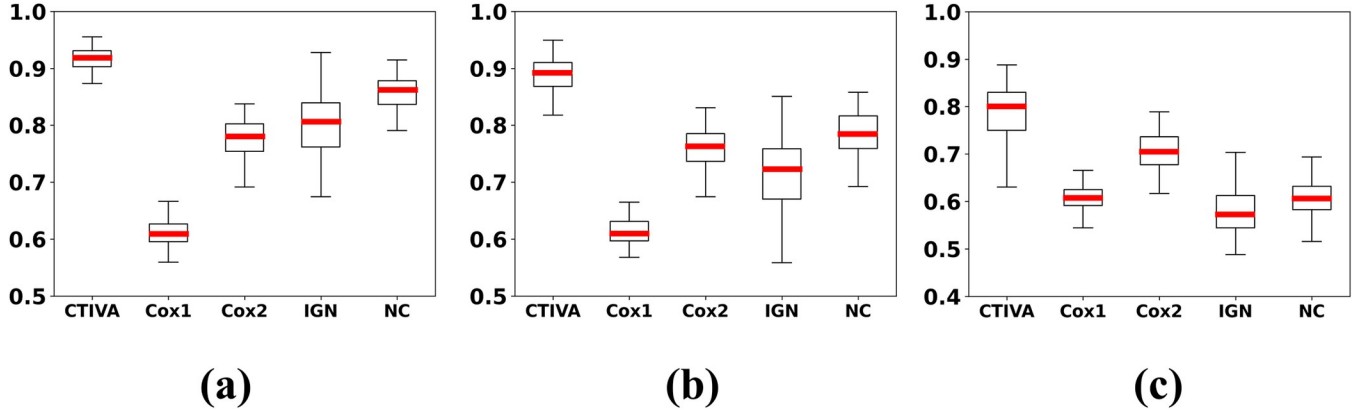

**Fig 2. Box plot of AUC in predicting combined variables with ANOVA test.** The CTIVA indicates the proposed method and other baseline methods in different datasets. The Cox1 and Cox2 describes the AUC results of Cox-survival analysis model each applied in censored time event 1 and 2. The IGN denotes the baseline methods that ignores the censoring and the NC denotes the baseline methods that only handles the non-censored data as the Fig 2. The data were sampled from additive exponential distributions where the power of the exponential value of the distribution were (a) 0.33, (b) 0.5, and (c) 1.

## Simulation results in correlated dataset

Although the proposed method showed promising results in uncorrelated simulation dataset, the time-interval correlated variables in real-world such as gene expression data are usually correlated by themselves. Therefore, the proposed method should also include the novel result in self-correlated dataset to show the effectiveness of the method in the real-world problems. The self-correlated datasets used in the experiment were generated through following process. Among the 1,000 categorical variables generated through the mentioned procedure, 10 groups each composed of 20 variables were randomly selected. The variables in the same group were correlated by adding the same white gaussian noise. Same as in independent categorical dataset and combined dataset identical statistical tests performed in categorical dataset were executed in the self-correlated dataset. Both the ROC curve and AUC values are acquired to demonstrate the novelty of the method as we have conducted in uncorrelated simulation setting.

The proposed CTIVA method also shows excellent prediction performance in the correlated categorical dataset. The performance of the CTIVA is compared with same baseline methods as we have conducted in the uncorrelated simulation setting. CTIVA exhibits average AUC values of 0.93, 0.94, and 0.93 in the case of categorical variables—with ANOVA, permutation test, and rank correlation test while the Cox survival analysis model and two naïve methods exhibit AUC values lower than 0.9. The average AUC values of CTIVA are observed to be 4.5, 5.6, and 8.1% higher than those of the most competitive alternative considered in the experiment on categorical dataset. The detailed comparison results are presented in Table 3.

Also, an additional experiment performed on the proposed method in the case of a real-world problem is suggested in the next sections to validate the effectiveness of the proposed method.

## Real dataset results

The results presented in the previous subsection demonstrate that the proposed CTIVA method exhibits novel performance on simulation problems. In this subsection, its performance is evaluated on the *National Sample Cohort Demo* (NSCD) dataset, which is a real-world medical history demo dataset provided by the *National Health Insurance Sharing Service* (NHISS) [21, 22].

NSCD contains the medical history of 1,000 randomly selected Korean healthcare beneficiaries sampled between 2002 and 2015. The proposed method is applied to the NSCD dataset to identify significant medical record variables correlated with the time interval between the diagnosis of *carcinoma* and death. Five medical record variables—*FORM_CD*, *MCARE_-SUBJ_CD*, *OFIJ_TYPE*, *OPRTN_YN*, and *MCARE_RSLT_TYPE*—are extracted from the healthcare statement data to test the correlation between the medical record variables and the censored time interval. Of these, *MCARE_SUBJ_CD* and *MCARE_RSLT_TYPE* are estimated

**Table 3. The AUC comparison results of proposed CTIVA and other baseline models in correlated dataset.**

| Method \ Test | ANOVA (+Linear) | Permutation Test | Rank Correlation Test |
|---|---|---|---|
| CTIVA | 0.93±0.03 | 0.94±0.03 | 0.93±0.03 |
| Cox Event 1 | 0.61±0.03 | | |
| Cox Event 2 | 0.82±0.03 | | |
| Ignored | 0.82±0.06 | 0.82±0.06 | 0.78±0.06 |
| No Censor | 0.89±0.02 | 0.89±0.02 | 0.86±0.03 |

The dataset was sampled from an additive exponential distribution with different statistical tests.

**Table 4. Description of NSCD Dataset and CTIVA-ANOVA analysis results.**

| Variable | FORM_CD | MCARE_SUBJ_CD | OFIJ_TYPE | OPRTN_YN | MCARE_RSLT_TYPE |
|---|---|---|---|---|---|
| **Variable Explanation** | Statement Manual Code | Medical Diagnosis Code | Beneficiary Division Code | Operation Progress Code | Medical Care Result Code |
| **CTIVA-ANOVA analysis p-value** | $7.08 \times 10^{-2}$ | $1.04 \times 10^{-4}$ | $5.72 \times 10^{-1}$ | $9.53 \times 10^{-1}$ | $1.30 \times 10^{-5}$ |

The detailed explanation of the extracted medical record variables of NSCD dataset and the description of the CTIVA-ANOVA test result analysis on the medical record variables.

to be correlated with the interval time when tested using CTIVA-ANOVA because the p-values of the two records are observed to be lower than 0.05. A detailed explanation of the medical record variables and descriptions of the test results are presented in Table 4.

When the diagnosis of *carcinoma* is considered to be event 1 and death of the beneficiary is considered to be event 2, the Medical Diagnosis Code (*MCARE_SUBJ_CD*) and Medical Care Result Code (*MCARE_RSLT_TYPE*) are estimated to be statistically significant corresponding to the interval time. The *MCARE_SUBJ_CD* and *MCARE_RSLT_TYPE* codes seem to be correlated with both events 1 and 2 because they are the diagnosis and result codes of the beneficiary during the visitation [23–25].

Furthermore, the proposed CTIVA is tested with proteasome inhibitor bortezomib dataset which is open access dataset provided by National Center for Biotechnology Information (NCBI) [26]. This dataset demonstrates the correlation between the outcome in clinical trials of the proteasome inhibitor bortezomib patients and gene expression profiling. Among tremendous gene expression data and medical categorical variables in the dataset, CTIVA is conducted to find the correlation between medical categorical variables and time interval between the pharmacogenomics (PGx) progression date and death. Since CTIVA covers both categorical and continuous variables, we have concentrated on medical categorical variable which censored data survival analysis usually does not cover.

Four medical categorical variables—age, sex, PGx response type, number of prior lines—are sampled from the proteasome inhibitor bortezomib dataset. The age of the patient is categorized under 65 or not to check whether the patient is in senescence. The sex of the patient was categorized as male or female. The PGx response type of the patient was categorized in six contents in dataset. The number of prior lines indicates the number of distinct treatments a patient has received for the cancer which was ranged from one to four. Among these medical categorical variables, the number of prior lines is known to be directly related with the survival time of the proteasome inhibitor bortezomib patients which indicates the correlation between the number of prior lines and interval time between PGx progression date and death [27]. The proposed CTIVA has found the number of prior lines to be statistically significantly correlated with the interval time in p-value 0.001 with ANOVA test. Other three variables are demonstrated to be not correlated showing p-value higher than 0.05 which corresponds with traditional studies [27]. The graphical description of the above experimental result is depicted in Fig 3. From the observed censored dataset, the proposed CTIVA estimates the actual event time and through the ANOVA test captures the correlated variables. Data which both of the events are censored is shown in Fig 3(A). The box plot in Fig 3(B) and 3(C) demonstrates the values of the interval time of the CTIVA estimated data and raw censored data with both events censored which is demonstrated as scatter plot in Fig 3(A). The horizontal label of the box plots indicates the variable number of prior lines that are known to be correlated with the interval time. The results shown in Fig 3 indicates the patient with less number of lines tend to

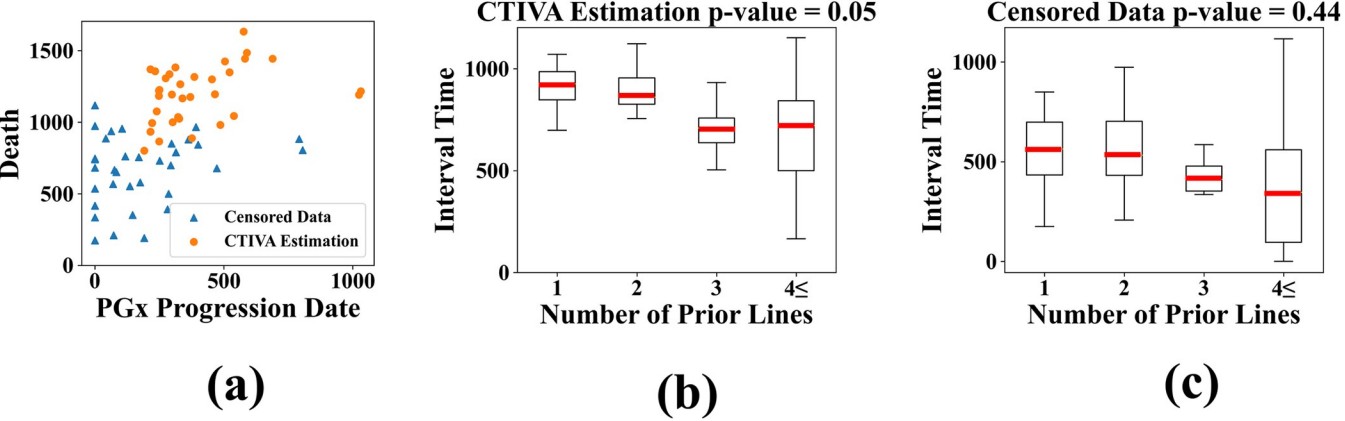

**Fig 3. Graphical description of CTIVA with NCBI dataset.** Observations which both PGx progression date and death are censored are used in the experiments. The triangle mark in (a) indicates raw censored data while dot mark indicates data estimated with CTIVA. The p-values acquired through ANOVA test in raw censored data and CTIVA estimated data are described in the title of (b) and (c). The interval times between PGx progression date and fatality for each number of prior lines are demonstrated in box plot (b) and (c).

have longer interval time between events which corresponds with the traditional studies [27]. The overall results of proteasome inhibitor bortezomib dataset can be acquired by running the source code provided in the github link below. [https://github.com/Insoo-K/CTIVA]

## Discussion and conclusion

Overall, the proposed CTIVA method exhibits excellent detection performance for variables that are correlated with the interval time of two events on both simulation data and real-world problems. In particular, the proposed method outperforms traditional statistical methods in capturing correlated variables while multiple censoring is presence. Implementing probability density function estimation and time interval estimation with Monte Carlo Estimation, the CTIVA exhibits an average AUC value exceeding 0.93 on simulation data by which outperforms conventional benchmark methods. Also, the CTIVA demonstrated its novel performance with real datasets—NSCD and NCBI datasets, in estimating categorical variables correlated with the interval time of two clinical events.

Further, CTIVA effectively estimated correlated variables on a combined dataset comprising both categorical and continuous variables. Although the use of identical p-values acquired via different statistical tests on categorical and continuous variables might not be mathematical justified, the results of the proposed method on simulated data are superior to those of the naïve methods, exhibiting an average AUC value of 0.90 in capturing correlated variables. Thus, the experimental results corroborate the effectiveness of such an approach in handling combined variable censored time-to-event datasets.

Moreover, CTIVA exhibits excellent detection of correlated variables on a real-world medical history dataset and clinical dataset with gene expression analysis. Among the five categorical medical record variables in the NSCD medical history dataset, the proposed method detects two variables correlated to the interval time between diagnosis of *carcinoma* and fatality. The two detected variables are clinically known to be correlated with the time interval between the two events. Additionally, among four categorical variables in NCBI datasets, the CTIVA captured prior lines to be correlated with the interval time between PG date and fatality. The number of prior lines of the patient is clinically shown to be correlated with the interval time of two events. Similar conclusions are also demonstrated in Fig 3 which indicates the patient with less number of lines tend to survive longer.

Although the CTIVA showed novel results in detecting correlated variables, some limitations exist in the proposed method. Since the proposed method estimates the joint probability distribution with the Bayesian estimation, disparity between the actual distribution and estimated distribution that influences the prediction results might exist. Also, the probability estimation of the proposed CTIVA relies on the convergence of the density estimation function. Therefore, computational cost of the proposed method might surge for the more accurate estimation of censored data compared to conventional benchmark methods.

In conclusion, the proposed CTIVA method is verified to be capable of effectively handling combined multivariate censored time-to-event datasets, which is the dominant type of censored dataset that appears in real-world problems. Despite some inherent limitations of the proposed method, we believe that the development of CTIVA is a milestone in research on handling combined multivariate censored time-to-event problems owing to its excellent performance on both simulation data and real-world problems. Further, we believe our milestone could contribute to real-world problems handling multivariate censored time-to-event problems with multiple censorings such as the expected survival time of carcinoma patients.

## Supporting information

**S1 Table. Comparison results of sensitivity and specificity at p-value 0.05.**
(DOCX)

**S2 Table. Comparison results of sensitivity and specificity at p-value 0.01 and p-value 0.1.**
(DOCX)

## Author Contributions

**Conceptualization:** Insoo Kim, Junhee Seok, Yoojoong Kim.

**Data curation:** Insoo Kim, Junhee Seok, Yoojoong Kim.

**Formal analysis:** Junhee Seok, Yoojoong Kim.

**Funding acquisition:** Yoojoong Kim.

**Investigation:** Insoo Kim, Junhee Seok, Yoojoong Kim.

**Methodology:** Insoo Kim, Junhee Seok, Yoojoong Kim.

**Project administration:** Insoo Kim, Junhee Seok, Yoojoong Kim.

**Resources:** Insoo Kim, Yoojoong Kim.

**Software:** Insoo Kim, Yoojoong Kim.

**Supervision:** Insoo Kim, Yoojoong Kim.

**Validation:** Insoo Kim, Junhee Seok, Yoojoong Kim.

**Visualization:** Insoo Kim, Yoojoong Kim.

**Writing – original draft:** Insoo Kim, Junhee Seok, Yoojoong Kim.

**Writing – review & editing:** Junhee Seok, Yoojoong Kim.

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
