## [Decision Letter · Decision Letter 0]

1 Aug 2023

PONE-D-23-20762CTIVA: Censored Time Interval Variable AnalysisPLOS ONE

Dear Dr. Kim,

Thank you for submitting your manuscript to PLOS ONE. After careful consideration, we feel that it has merit but does not fully meet PLOS ONE’s publication criteria as it currently stands. Therefore, we invite you to submit a revised version of the manuscript that addresses the points raised during the review process.

We look forward to receiving your revised manuscript.

Kind regards,

Guanghui Liu

Academic Editor

PLOS ONE

“This research was supported by a grant of the Basic Science Research Program through the National Research Foundation of Korea (NRF) funded by the Ministry of Education (NRF-2021R1I1A1A01044255) and was supported by the Research Fund, 2022 of The Catholic University of Korea.**”**

“YooJoong Kim has recieved funds in below.

This research was supported by a grant of the Basic Science Research Program through the National Research Foundation of Korea (NRF, https://www.nrf.re.kr/) funded by the Ministry of Education (NRF-2021R1I1A1A01044255) and was supported by the Research Fund, 2022 of The Catholic University of Korea (https://www.catholic.ac.kr/english/).

Reviewers' comments:

Reviewer's Responses to Questions

**Comments to the Author**

1. Is the manuscript technically sound, and do the data support the conclusions?

Reviewer #1: Yes

Reviewer #2: Partly

Reviewer #3: Partly

2. Has the statistical analysis been performed appropriately and rigorously? 

Reviewer #1: Yes

Reviewer #2: No

Reviewer #3: Yes

3. Have the authors made all data underlying the findings in their manuscript fully available?

Reviewer #1: Yes

Reviewer #2: No

Reviewer #3: Yes

4. Is the manuscript presented in an intelligible fashion and written in standard English?

Reviewer #1: Yes

Reviewer #2: Yes

Reviewer #3: Yes

5. Review Comments to the Author

Reviewer #1: Overall the paper proposed a novel way to handle multiple censored datasets. The paper would be stronger if CTIVA can be compared with other state of the art methods such as survival analysis techniques or other machine learning methods.

While the paper applies CTIVA to a real-world medical history dataset, the analysis is limited to a small subset of variables. It would be beneficial to include a more comprehensive analysis on larger real-world datasets to demonstrate the generalizability and robustness of CTIVA in different domains and data types.

Reviewer #2: The authors proposed CTIVA method and showed its capability in handling combined multivariate censored time-to-event datasets, which demonstrated exceeding analytical power than methods that ignore the censoring or only handles the non-censored data. Please find the comments below

1. The concept of the work was clearly demonstrated but the data to support the concept was notably lacking. For example, the AUC was calculated and compared among multiple algorithms but explanation on how the data was derived was not clear and the statistic significance was missing. For example in Figure 1 and 2, what is the cutoff when deriving the AUC; what is statistical significance of each AUC; what is the sensitivity and specificity of each AUC; what is the significance when comparing three algorithms? It is suggested for authors to include more data in the supplementary information.

2. Certainly, as stated in the Data Availability section, there was a link. However, it requests a fee to access the data. This lack of access hindered my ability to fully comprehend and evaluate the authenticity of this study. The lack of data availability raises concerns about the reproducibility and reliability of the study's results.

3. The introduction is concise and clear. However, it may pose confusion for readers who are not familiar in this area. It is suggested for authors to add one more paragraph at Introduction section on the impact of censored data analysis in real-world cases and expand a bit on those statistic tests you used in comparing CTIVA to IGN and NC.

Reviewer #3: 1. The background provided in this work could benefit from a more comprehensive and specific review of related references. Distinguishing CTIVA from other existing methodologies, such as the competing risk regression model, would enhance the clarity and uniqueness of the approach.

2. To improve the manuscript, the authors should include more detailed information in key sections. For instance, providing a thorough sensitivity analysis of different parameter settings in the simulation study would strengthen the validity of the findings. Additionally, clear descriptions of baseline models compared to CTIVA and the rationale behind their selection for comparison should be included.

3. As a recommendation, appending the scripts or source codes used in the study would be valuable. This would allow future researchers to replicate the results and verify the findings when necessary. Furthermore, it is essential to assess whether this methodology can be adopted by others in the field. Publishing the methodology as an R package or making it accessible for future researchers would be beneficial for broader application and dissemination.

6. PLOS authors have the option to publish the peer review history of their article (what does this mean?). If published, this will include your full peer review and any attached files.

Reviewer #1: No

Reviewer #2: No

Reviewer #3: No

---

## [Author Response · Author response to Decision Letter 0]

18 Sep 2023

Reviewer #1:

Comments 1. Overall, the paper proposed a novel way to handle multiple censored datasets. The paper would be stronger if CTIVA can be compared with other state of the art methods such as survival analysis techniques or other machine learning methods. 

Author’s response: We would like to express our appreciation for the valuable feedback provided by the reviewer. We have carefully considered their comments and have incorporated their suggestions into our manuscript. 

Regarding the reviewer’s suggestion to compare the proposed CTIVA model with other state of art methods, we agree that this would help demonstrating the effectiveness of the proposed method. Therefore, we have added a comparison result of CTIVA model and Cox survival analysis model in the simulation result section.

As highlighted in the revised manuscript, the additional test results acquired through the Cox survival analysis is provided in the simulation result section. We have modified the Table 2 and 3 and Figure 1 and 2 to add the Cox survival analysis results [1]. We believe supplementing the data with the existing state of the art models would help emphasizing the effectiveness of the proposed method. 

Comments 2. While the paper applies CTIVA to a real-world medical history dataset, the analysis is limited to a small subset of variables. It would be beneficial to include a more comprehensive analysis on larger real-world datasets to demonstrate the generalizability and robustness of CTIVA in different domains and data types.

Author’s response: Thank you for your valuable comment on our paper. We appreciate your suggestion to extend our method on larger real-world datasets. We have applied CTIVA in proteasome inhibitor bortezomib dataset provided by National Center for Biotechnology Information (NCBI) [2]. The application result of the proposed CTIVA in proteasome inhibitor bortezomib dataset is supplemented in Real Dataset Results section. The proposed CTIVA also demonstrated novel performance in new dataset finding the medical categorical variable; number of prior lines to be statistically significantly correlated with the time interval between pharmacogenomics progression date and death. The result of the CTIVA corresponds with the traditional medical studies about clinical trial outcome of proteasome inhibitor bortezomib patients [3].

Reviewer #2:

Comments 1. The concept of the work was clearly demonstrated but the data to support the concept was notably lacking. For example, the AUC was calculated and compared among multiple algorithms but explanation on how the data was derived was not clear and the statistic significance was missing. For example in Figure 1 and 2, what is the cutoff when deriving the AUC; what is statistical significance of each AUC; what is the sensitivity and specificity of each AUC; what is the significance when comparing three algorithms? It is suggested for authors to include more data in the supplementary information.

Author’s response: We appreciate the reviewer’s helpful comments in improving the manuscript. We have carefully considered their comments and have implemented their suggestion into our manuscript. 

Regarding the reviewer’s suggestion to enhance the description about the AUC in simulation result section, we additionally provided the cutoff while driving the AUC; sensitivity and specificity of each AUC in the simulation result section. In specific, the cutoff value of AUC was set as 0.05 and the CTIVA showed superior results compared to baseline method in both sensitivity and specificity with the provided cutoff value. With the cutoff p-value 0.05 in categorical dataset, the average sensitivity of CTIVA is 0.96 while the average specificity is 0.84. Additionally, in combined dataset, the average sensitivity of CTIVA is 0.88 while the mean specificity is 0.83. These modifications are added in page 11, Simulation Results in Uncorrelated Dataset section and we have included detailed comparison in supplementary. 

Comments 2. Certainly, as stated in the Data Availability section, there was a link. However, it requests a fee to access the data. This lack of access hindered my ability to fully comprehend and evaluate the authenticity of this study. The lack of data availability raises concerns about the reproducibility and reliability of the study's results.

Author’s response: Thank you for your acute point that data is not available. We have found out that demo data we used for the real-world test is only freely available for domestic users who are citizens of South Korea. The Cohort Demo dataset we have used for the experiment is open access for the domestic users but not freely available for foreign users and require fee for the access. We deeply apologize the inconvenience we have made due to the lack of double checking the data sharing policy of National Health Insurance Sharing Service of Republic of Korea for foreign users.

 Due to the lack of data availability of the Cohort Demo dataset, we have committed proteasome inhibitor bortezomib dataset provided by NCBI in github. The source code that generates the simulation samples and applies the CTIVA with the provided dataset is also committed in the following link [https://github.com/Insoo-K/CTIVA]. 

Comments 3. The background provided in this work could benefit from a more comprehensive and specific review of related references. Distinguishing CTIVA from other existing methodologies, such as the competing risk regression model, would enhance the clarity and uniqueness of the approach.

Author’s response: We gradually appreciate the reviewer’s helpful suggestion in improving the manuscript by supplementing the related references in the manuscript. We have implemented the reviewer’s suggestion in the introduction section and inserted the paragraph in the manuscript as below. 

Although, there exist survival analysis models that handle multiple events such as competing risk regression model, it only handles the correlated events. Therefore, method that handles multiple independent censored events, should be further investigated. 

The proposed CTIVA handles multiple independent censored time to event data where competing risk regression model cannot be implemented.

Reviewer #3:

Comments 1. The background provided in this work could benefit from a more comprehensive and specific review of related references. Distinguishing CTIVA from other existing methodologies, such as the competing risk regression model, would enhance the clarity and uniqueness of the approach. 

Author’s response: We gradually appreciate the reviewer’s valuable suggestion in improving the manuscript by demonstrating the uniqueness of CTIVA compared to existing methods. Our proposed CTIVA handles multiple independent censored time to event data where most of existing methods cannot handle. We have applied the reviewer’s suggestion in the introduction section and modified the manuscript as below. 

Although, there exist survival analysis models that handle multiple events such as competing risk regression model, it only handles the correlated events. Therefore, method that handles multiple independent censored events, should be further investigated. 

Comments 2. To improve the manuscript, the authors should include more detailed information in key sections. For instance, providing a thorough sensitivity analysis of different parameter settings in the simulation study would strengthen the validity of the findings. Additionally, clear descriptions of baseline models compared to CTIVA and the rationale behind their selection for comparison should be included.

Author’s response: We thank the reviewer’s helpful suggestion in enriching the manuscript. We have implemented the reviewer’s valuable feedback in the manuscript and inserted the additional sensitivity and specificity analysis in Simulation Results in Uncorrelated Dataset section and Supplementary. Also, we have added descriptions of baseline models in the Simulation Settings section as below. 

The proposed CTIVA method was compared with various baseline methods for the performance verification. The baseline methods include naïve methods that do not consider the presence of censoring by ignoring the censoring or only considering the sample without censoring. In specific, the naïve method that ignores censoring considers the censored time event as actual observed event. Furthermore, we have also conducted the experiment with Cox Regression Model, the method that is dominantly used for handling the censored time to event data [4]. 

1. Cox DR. Regression models and life‐tables. Journal of the Royal Statistical Society: Series B (Methodological). 1972;34(2):187-202.

2. Mulligan G, Mitsiades C, Bryant B, Zhan F, Chng WJ, Roels S, et al. Gene expression profiling and correlation with outcome in clinical trials of the proteasome inhibitor bortezomib. Blood. 2007;109(8):3177-88.

3. Boccadoro M, Morgan G, Cavenagh J. Preclinical evaluation of the proteasome inhibitor bortezomib in cancer therapy. Cancer cell international. 2005;5:1-9.

4. Therneau TM, Grambsch PM. The cox model. Modeling survival data: extending the Cox model: Springer; 2000. p. 39-77.

---

## [Decision Letter · Decision Letter 1]

3 Oct 2023

PONE-D-23-20762R1CTIVA: Censored Time Interval Variable AnalysisPLOS ONE

Dear Dr. Kim,

Thank you for submitting your manuscript to PLOS ONE. After careful consideration, we feel that it has merit but does not fully meet PLOS ONE’s publication criteria as it currently stands. Therefore, we invite you to submit a revised version of the manuscript that addresses the points raised during the review process.

We look forward to receiving your revised manuscript.

Kind regards,

Guanghui Liu

Academic Editor

PLOS ONE

Journal Requirements:

Reviewers' comments:

Reviewer's Responses to Questions

**Comments to the Author**

1. If the authors have adequately addressed your comments raised in a previous round of review and you feel that this manuscript is now acceptable for publication, you may indicate that here to bypass the “Comments to the Author” section, enter your conflict of interest statement in the “Confidential to Editor” section, and submit your "Accept" recommendation.

Reviewer #2: All comments have been addressed

Reviewer #3: (No Response)

Reviewer #4: (No Response)

2. Is the manuscript technically sound, and do the data support the conclusions?

Reviewer #2: Yes

Reviewer #3: (No Response)

Reviewer #4: Yes

3. Has the statistical analysis been performed appropriately and rigorously? 

Reviewer #2: Yes

Reviewer #3: (No Response)

Reviewer #4: Yes

4. Have the authors made all data underlying the findings in their manuscript fully available?

Reviewer #2: Yes

Reviewer #3: (No Response)

Reviewer #4: No

5. Is the manuscript presented in an intelligible fashion and written in standard English?

Reviewer #2: Yes

Reviewer #3: (No Response)

Reviewer #4: Yes

6. Review Comments to the Author

Reviewer #2: (No Response)

Reviewer #3: Thank you for addressing my comments.

1. Tables 2 and 3 lack clear descriptions. In the context of sensitivity analysis, it would be helpful to know if there were any disparities in the results when various parameter hypotheses were considered. Additionally, some details about the simulation process, such as the distribution characteristics of the variables, would be valuable for a more comprehensive understanding. We anticipate that CTIVA will outperform the baseline methods across various parameter hypotheses and settings.

2. I would suggest the authors incorporate a more in-depth discussion regarding CTIVA, including the inherent limitations previously mentioned in the discussion section.

Reviewer #4: The manuscript introduces a novel methodology, CTIVA, designed for the analysis of datasets characterized by multiple censored time-to-events. The exposition is lucid, and the narrative is well-structured, making it accessible to readers. The quality of manuscript might be further enhanced when addressing the following issues:

1.The work titled "Network estimation for censored time-to-event data for multiple events based on multivariate survival analysis" appears to be closely aligned with the themes of this manuscript. It would enhance the manuscript's depth if this work is introduced early on, preferably in the introduction. A comparative analysis with other existing methodologies, highlighting the advancements and distinctions of CTIVA in analyzing censored time-to-event data, would provide readers with a clearer understanding of the manuscript's contributions.

2. While the manuscript provides a comprehensive list of references, there appears to be a reliance on older publications. Specifically, only 4 out of the 21 cited papers have been published in the last four years, with the most recent one dating back to 2021. To ensure the manuscript is grounded in the latest research, it would be beneficial to incorporate more contemporary references relevant to the topic.

3. To further elucidate the results and underscore the advantages of CTIVA, it is suggested incorporating more visual demonstrations, particularly when juxtaposing CTIVA with other methods. Graphical representations can offer intuitive insights and make the comparisons more tangible for readers.

7. PLOS authors have the option to publish the peer review history of their article (what does this mean?). If published, this will include your full peer review and any attached files.

Reviewer #2: **Yes: **Rongwei Lei

Reviewer #3: No

Reviewer #4: No

---

## [Author Response · Author response to Decision Letter 1]

30 Oct 2023

Reviewer #3:

Comments 1. Tables 2 and 3 lack clear descriptions. In the context of sensitivity analysis, it would be helpful to know if there were any disparities in the results when various parameter hypotheses were considered. Additionally, some details about the simulation process, such as the distribution characteristics of the variables, would be valuable for a more comprehensive understanding. We anticipate that CTIVA will outperform the baseline methods across various parameter hypotheses and settings.

Author’s response: We greatly appreciate your constructive suggestions, which have significantly contributed to enhancing our manuscript. We have conducted an additional experiment verifying the effectiveness of CTIVA in various setting. Varying the cutoff p-value by 0.1 and 0.01, the proposed CTIVA still outperformed other benchmark methods in terms of sensitivity and specificity. Additionally, we have revised some value errors overlooked in Table S1. The detailed comparison results are demonstrated in Table S1 and S2 in Supplementary section. Furthermore, we have refined the manuscript by moving the Table 1, which analyzes data generation procedure, to Simulation Setting section while including the detailed explanation about datasets generated through simulation right behind the Table 1.

Comments 2. I would suggest the authors incorporate a more in-depth discussion regarding CTIVA, including the inherent limitations previously mentioned in the discussion section.

Author’s response: Thank you for your valuable comment on our paper. We appreciate your suggestion to supplement the inherent limitations of the proposed CTIVA in the discussion section of the manuscript. The limitations of the proposed method due the joint probability function estimation are described in the manuscript as written below. 

Although, the CTIVA showed novel results in detecting correlated variables, some limitations exist in the proposed method. Since the proposed methods estimates the joint probability distribution with the Bayesian estimation, disparity between the actual distribution and estimated distribution that influences the prediction results might exist. Also, the proposed CTIVA relies the probability estimation on the convergence of the density estimation function. Therefore, due to the variety of the computing time of the function convergence, the computational cost of the proposed method might surge depending on the task. 

Reviewer #4:

Comments 1. The work titled "Network estimation for censored time-to-event data for multiple events based on multivariate survival analysis" appears to be closely aligned with the themes of this manuscript. It would enhance the manuscript's depth if this work is introduced early on, preferably in the introduction. A comparative analysis with other existing methodologies, highlighting the advancements and distinctions of CTIVA in analyzing censored time-to-event data, would provide readers with a clearer understanding of the manuscript's contributions.

Author’s response: We greatly appreciate the reviewer’s valuable suggestion supplementing the manuscript by referring the previous study about multivariate survival analysis. We have introduced the previous study covering multivariate censored data in introduction section. The inserted paragraph is highlighted in manuscript with tracked changes as yellow. Furthermore, we have demonstrated the distinct advantage of the proposed CTIVA in handling categorical and combined variables. Specifically, we have inserted the following paragraph in the last paragraph of introduction section to notate the contribution of the CTIVA. 

Moreover, CTIVA demonstrates practical advantages compared to previous methods. Our proposed method addresses challenges posed by diverse variable types, a facet not adequately tackled by prior studies. Specifically, the proposed method could be implemented in clinical trial and symptoms which could be considered as categorical variables. Thus, the proposed CTIVA has extended the solution of multivariate censored time-to-event dataset to more general fields.

Comments 2. While the manuscript provides a comprehensive list of references, there appears to be a reliance on older publications. Specifically, only 4 out of the 21 cited papers have been published in the last four years, with the most recent one dating back to 2021. To ensure the manuscript is grounded in the latest research, it would be beneficial to incorporate more contemporary references relevant to the topic.

Author’s response: We thank the reviewer’s helpful suggestion to enhance the manuscript by citing more contemporary references. We have reviewed recent literatures that highlight cutting-edge method for handling censored data in survival analysis. In particular, we have cited four latest research papers related to our work in Introduction section and cited a latest research paper about survival analysis in Results section. The cited papers in introduction section demonstrate the arising significance of handling multivariate censored time-to-event data while the paper cited in result section indicates the varying result of survival analysis attributed to diverse distributions which our research has handled.

Comments 3. To further elucidate the results and underscore the advantages of CTIVA, it is suggested incorporating more visual demonstrations, particularly when juxtaposing CTIVA with other methods. Graphical representations can offer intuitive insights and make the comparisons more tangible for readers. 

Author’s response: We thank the reviewer’s valuable comments to refine the manuscript by incorporating visual demonstration of the proposed CTIVA. We have included the graphical comparison of the raw censored data and data estimated with CTIVA in NCBI dataset with Fig 3 in result section. In the Fig 3, subfigure (a) demonstrates the difference between raw censored data where both event 1 and 2 are censored and data estimated with CTIVA. Additionally, subfigure (b) depicts the ANOVA result of the raw censored data and estimated data with bar graph of average values of interval time. The p-values provided in subfigure (b) are calculated through the ANOVA test.

---

## [Editor Report · Decision Letter 2]

3 Nov 2023

CTIVA: Censored Time Interval Variable Analysis

PONE-D-23-20762R2

Dear Dr. Kim,

We’re pleased to inform you that your manuscript has been judged scientifically suitable for publication and will be formally accepted for publication once it meets all outstanding technical requirements.

Kind regards,

Guanghui Liu

Academic Editor

PLOS ONE
---

## [Editor Report · Acceptance letter]

7 Nov 2023

PONE-D-23-20762R2 

CTIVA: Censored Time Interval Variable Analysis 

Dear Dr. Kim:

I'm pleased to inform you that your manuscript has been deemed suitable for publication in PLOS ONE. Congratulations! Your manuscript is now with our production department. 

Kind regards, 

on behalf of

Dr. Guanghui Liu 

Academic Editor

PLOS ONE